# Colibactin possessing *E. coli* isolates in association with colorectal cancer and their genetic diversity among Pakistani population

**Habiba Tariq**[1], **Zobia Noreen**[1], **Aftab Ahmad**[1,2], **Laraib Khan**[1], **Mashhood Ali**[3], **Muhammad Malik**[1], **Aneela Javed**[4], **Faisal Rasheed**[5], **Alina Fatima**[1], **Tanil Kocagoz**[6], **Ugur Sezerman**[7], **Habib Bokhari**[1,2]*

1 Department of Biosciences, COMSATS University, Islamabad, Pakistan, 2 Department of Microbiology, Kohsar University Murree, Punjab, Pakistan, 3 Department of Gastroenterology, Pakistan Institute of Medical Sciences (PIMS), Islamabad, Pakistan, 4 Atta-ur-Rahman School of Applied Biosciences (ASAB), National University of Science and Technology (NUST), Islamabad, Pakistan, 5 Department of Microbiology, Quaid-e-Azam University, Islamabad, Pakistan, 6 Department of Medical Microbiology, Acibadem University, Istanbul, Turkey, 7 Department of Biostatics and Medical Informatics, Acibadem University, Istanbul Turkey

* habib@comsats.edu.pk, vc@kum.edu.pk

**Data Availability Statement:** All relevant data are within the paper and its Supporting Information files.

## Abstract

Colorectal cancer (CRC) is the third most prevalent cause of tumorigenesis and several pathogenic bacteria have been correlated with aggressive cases of cancer i.e., genotoxin (colibactin) producing *Escherichia coli* (*E. coli*). This study was designed to investigate the genetic diversity of clb⁺clb⁺ *E. coli* strains and their association with CRC. Pathogenic *E. coli* isolates from colorectal biopsies were characterized based on phylotypes, antibiotic resistance pattern, and (Enterobacterial Repetitive Intergenic Consensus Sequence-based Polymerase Chain Reaction) ERIC-PCR. Furthermore, isolates were screened for the presence of the *Pks* (polyketide synthase) *Island* specifically targeting colibactin genes A and Q. The selective clb⁺clb⁺ isolates were subjected to cytotoxicity assay using Human embryonic kidney (HEK) cell lines. We revealed that 43.47% of the cancer-associated *E. coli* isolates were from phylogroup B2 comparatively more pathogenic than rest while in the case of healthy controls no isolate was found from B2. Moreover, 90% were found positive for colibactin and *pks* (polyketide synthase) island, while none of the healthy controls were found positive for colibactin genes. All healthy and cancer-associated isolates were tested against 15 antibiotic agents, we observed that cancer-associated isolates showed a wide range of resistance from 96% against Nalidixic acid to 48% against Doxycycline. Moreover, *E. coli* isolates were further genotyped using ERIC-PCR, and selected clb⁺clb⁺ *E. coli* isolates were subjected to cytotoxicity assay. We recorded the significant cytotoxic activity of clb+clb+ *E. coli* phylogroup B2 isolates that might have contributed towards the progression of CRC or dysbiosis of healthy gut microbiota protecting against CRC pathogenesis. Our results revealed a significant $p<0.023$ association of dietary habits and hygiene $p<0.001$ with CRC. This is the first study to report the prevalence of *E. coli* phylogroups and the role of colibactin most virulent phylogroup B2 among Pakistani individuals from low socioeconomic setup.

**Funding:** The research work was funded by PROJECT NO.PSF-TUBITAK/C-COMSATS (01) entitled "Developing Protocols for multiplex PCR based Screening and Computational Models for Virulence Factors associated with H. pylori and colibactin (genotoxin) producing E. coli.

**Competing interests:** The authors have declared that no competing interests exist.

## Introduction

A large proportion of the world population is suffering from colorectal cancer (CRC) and it is recognized globally as the third leading cause of human cancer-related deaths [1]. Moreover, according to the World Health Organisation report, the rate of CRC is substantially higher among males in comparison to females [2]. In Pakistan, it is considered the third most common cancer among men and second among women with 10% and 9.2% respectively of total reported cases [3]. Most of the colorectal cancer cases are recognized as sporadic (~90%) in nature whereas only 5 to 10% cases of CRC have genetic association whereas, 1 to 2% cases are reported in the context of various inflammatory bowel diseases. Sporadic CRC cases are mainly influenced by various factors such as diet, environment, and intestinal microbiota [4].

Approximately, $10^{11}$ to $10^{12}$ bacterial cells per milliliter are residing in the human colon as intestinal microbiota making the colon the most densely populated microbial habitat on earth in terms of occupancy of microbes in a small area but also terms of total information content (encoding more than 3 million genes) [5]. These bacteria are distinguished into more than 1000 species that constitute a wider community or microbiota and out of this microbiota; it is natural to assume the association/abundance of a few bacteria with pathogenicity of human CRC (colorectal cancer). Moreover, through the action of various microbial gene products, metabolites, and multiple microbial structural components, this microbiota performs an essential role in regulating host immunity, metabolic activity, intestinal homeostasis, and gut barrier function. Microbial dysbiosis is mainly associated with various intestinal abnormalities such as Inflammatory Bowel Disease (IBD) and colorectal cancer [6]. In this regard, *E. coli* capable of adhering to intestinal epithelial cells also carries numerous toxins, named cyclomodulins that include colibactin, cytolethal distending toxin (CDT), cycle inhibiting factor, and cytotoxic necrotizing factor (CNF), have been reported from CRC patients. Particularly, a toxin named colibactin, synthesized from *pks* island was preferentially found among the strains isolated from CRC patients [7]. It has also been perceived in recent studies that the colonization of colonic mucosa by *E. coli* strains particularly possessing *pks* island and producing colibactin are involved in the progression of CRC, either by producing genotoxins or by disturbing the host's mismatch repair pathway (MMR) that results in microsatellite instability (MSI) [8].

*E. coli* strains are further categorized into four major phylogenetic groups i.e. A, B1, B2, and D [9]. Strains of *E. coli* that belong to different phylogenetic groups possess distinct properties such as antibiotic resistance profile, pathogenic potential, genome size, the capability of exploiting ecological niches and available sugars [10]. Cowling and Gordon in 2003 presented that the relative abundance of *E. coli* phylogroups depends majorly on the diet, body mass, and the surrounding environment of the host-specific niche. Phylogroups A and B2 in combination are mostly associated with humans, whereas phylogroups (A and B1) are commonly prevalent among non-human mammals, while the subgroup of phylogroups B2 has only been reported from humans [11]. Determination of colonic-mucosa-associated *E. coli* isolated from CRC patients conferred the highest percentage of cyclomodulin-producing *E. coli* belonging to B2 phylogroup and most of them possessed *pks* island that encodes colibactin and *cnf* genes [12]. In industrialized countries, far-reaching epidemiological statements about commensal *E. coli* populations reported the pervasiveness of groups A and B2, including clb+ strains. The distribution of the *E. coli* phylogroup depends on numerous variables such as diet, lifestyle, and hygiene status, with fluctuations in the ratio between phylogroup A and B2, while the rest are rarer [13].

Nougayrede and his companions, in 2006, identified a meningitis *E. coli* strain IHE3034. They identified the *polyketide synthase island* that produced a genotoxin named the toxin

colibactin [14]. Colibactin is a naturally occurring genotoxic chemical compound that is synthesized by non-ribosomal peptide synthases, polyketide synthases, and hybrid enzymes encoded by a 54 kb genomic island. This toxin works by inducing arrest in the cell cycle at G2 or M phase, chromosomal aberrations, and by causing a double-strand break in DNA [15]. Moreover, it has been reported that colibactin-harbouring *E. coli* are highly represented among colorectal cancer (CRC) patients and have shown an increased number of tumors in various colorectal cancer mouse models [16].

Current studies have suggested that some specified strains of *Escherichia coli* (*E. coli*) possess a cluster of genes named polyketide synthase (*pks*) island, which has been associated with the development of CRC in humans [17]. The *Pks Island* possesses a 54-kb genome size that encodes for the colibactin (clb) gene cluster. The colibactin is recognized as a genotoxic metabolite that acts as cyclomodulin which takes part in enhancing DNA damage and also leads to cell cycle arrest in mammalian cells. This polyketide synthase island has been proved to be highly conserved in *Enterobacteriaceae* and also has been isolated from B2 strains of *E. coli* [18].

*In vitro* studies have presented that *pks*+ *E. coli* strains promote megalocytosis i.e. cells and nuclei enlargement without mitosis, induce DNA double-strand breaks, and encourage G2 cell cycle arrest. Human embryonic kidney(HEK) cells have been widely used cytoxicity studies in cell biology research because of their reliable growth and transfection, experimental cytotoxicity assays using HEk cell culture presented that *pks*-encoding *E. coli* plays a major role in colonizing, causing inflammation and cancer progression [19].

It has been extensively reported that antibiotic resistance is the main cause of reduction in the effectiveness of drugs during the treatment of various diseases including colorectal cancer. Drug resistance develops in approximately all colon cancer patients and that leads to decreased therapeutic efficacies of anticancer agents [20]. In this communication, we studied the prevalence and association of clb+ *E. coli* isolate cultured from colorectal cancer patients' biopsies and healthy individuals. The isolated *E. coli* strains were then characterized based on antibiotic resistance profiling, phylogrouping, and ERIC PCR to further explore their correlation with different grades of CRC. Cytoxicity assay was performed on selected clb+ isolates using HEK Cell lines to study the cytotoxic effect of colibactin on growing cells.

## Materials and methods

### Samples collection

Patients suffering from colorectal cancer, declared by the in-charge physician and confirmed by histopathological reports, attending the department of Gastroenterology, Pakistan Institute of Medical Sciences (PIMS) Islamabad were enrolled in the study. Patients aged less than 20 years and suffering from severe gastrointestinal infection and bleeding were not included in this study. This research study was approved by ethical review board, PIMS (Pakistan Institute of Medical Sciences) Islamabad. Informed written consent was collected from enrolled patients. Biopsy samples were collected from enrolled patients along with controls and processed for culturing. Questionnaires were filled using the provided information by the patient to collect the socio-demographic data (S2 Table).

### Culturing and biochemical identification

A biopsy sample from each patient was collected, kept in 20% glucose solution, and transported to the laboratory on the ice. Biopsies were then homogenized and preparations were cultured on MacConkey agar plates and incubated at 37˚C for 18–24 hours for bacterial colonies isolation. Based on morphological characterization, single *E. coli* colonies were red-

streaked and purified on MacConkey agar/LB agar incubated at 37˚C overnight. The identification of *E. coli* colonies was confirmed by Gram's staining and various biochemical tests (i.e. Triple Sugar Iron (TSI), Citrate Test, Catalase, Oxidase, Indole Test, Methyl red test, and Urease test). Pure *E. coli* cultures confirmed by morphological and biochemical tests were processed and preserved in 20% glycerol and stored at -80˚C until further processing.

## DNA extraction and quantification

Genomic DNA from *E. coli* strains was extracted by ethanol precipitation method using the method described by [21]. Briefly, a single pure bacterial colony selected from each sample was resuspended in 20 μL of 1% sodium dodecyl sulphate, 40 μL proteinase K (100 μg/mL), 80 μL of proteinase K buffer (4M NaCl, 0.5M EDTA; pH 7.5), and incubated at 55˚C for 1 hour. Subsequently, 100 μL 6M NaCl was added, vortexed for 1 minute, and centrifuged at $16,863 \times g$ for 1 minute, at 4˚C. Nucleic acids were then precipitated by adding absolute ethanol and harvested by centrifugation ($16,863 \times g$ at 4˚C for 1 minute). Resultant DNA pellets were each washed with 70% vol/vol ethanol and then re-suspended in 100 μL 10 mM Tris, 1 mM EDTA buffer; pH 7.5. Samples were stored at −20˚C. The extracted genomic DNA was quantified using Nano-drop 2000c Colibri TITERTEK BRETHOLD and visualized by gel electrophoresis.

## Phylogrouping using Quadraplex PCR

Genomic DNA extracted from each *E. coli* isolate was further used for Phylogrouping by Quadruplex PCR using the method described by (Clermont *et al*., 2013). The PCR amplified products were analyzed on 2% agarose gel for the confirmation of the amplified targeted product. The primers used for the determination of phylogroups (*chu*A, *yja*A, *TspE4.C2, and arpA*) are listed in (S1 Table).

## PCR conditions for phylogrouping

The PCR conditions used for characterization of phylogroups were as follows: initial denaturation at 94˚C for 5minutes, denaturation at 94˚C for 1 minute, annealing at 59˚C for 1 minute, extension at 72˚C for 1 minute followed by the final extension at 72˚C for 7 minutes.

## Detection of colibactin genes

*E. coli* isolates belonging to B2 phylogroups were further investigated for genotoxin (colibactin) production. The primers used for the determination of clb+ *E. coli* isolates (*Clb*A and *Clb*Q) are shown in (S1 Table). The PCR conditions used for colibactin genes amplification were as the initial denaturation was given at 95˚C for 10 minutes, denaturation at 95˚C for 15 seconds, annealing at 60˚C for 1 minute, extension at 72˚C for 40 seconds followed by the final extension at 72˚C for 7 minutes.

## Antibiotics susceptibility profiling

The antibiotic susceptibility was determined by the standardized method described by the Kirby-Bauer disk diffusion method on Mueller-Hinton agar plates keeping CLSI 2017 recommendations as standards. The used antibiotic disks were Doxycycline (DO) (30 μg), Nalidixic acid (NA) (30 μg), Ceftazidime (CAZ) (30 μg), Erythromycin (E) (15 μg), Amoxicillin (AML) (20 μg), Gentamicin (CN) (10 μg), Trimethoprim-sulfamethoxazole (SXT) (25 μg), Rifampin (RD) (5 μg), Ampicillin (Amp) (10 μg), Amikacin (AK) (30 μg), Levofloxacin (LEV) (5 μg), Ofloxacin (OFX) (5 μg), Colistin (CT) (10 μg), Amoxicillin-clavulanic Acid (AMC) (20 μg) and Ciprofloxin (CIP) (5 μg). Antibiotic susceptibility was ascertained after 18–24 hours of

incubation at 37°C as described by (Velican et al., 2020) [22]. *E. coli* strains were classified as resistant, susceptible and intermediate by comparison of diameter of inhibition to the CLSI, 2017 guidelines.

### ERIC-PCR genotyping

To determine the evolutionary relationship among the extracted sixty 60 isolates, genotyping was performed using ERIC fingerprinting assay. In this assay, a 22bps primer designed against the specific and conserved ERIC region was used. The PCR was performed in a total volume of 25μl reaction mixture containing 100 ng of DNA and 25pmol of ERIC 2 primer (5-AAGTAAG TGACTGGGGTG AGCG-3) [23].

### Viability/cytotoxicity of bacterial culture against HEK cell lines

*E. coli* strains screened in the cytotoxicity assay included 5002 (colibactin negative, noncancerous source), H21C (a clb$^+$ control), 190, 364 (clb$^+$ CRC source of this study), and IHE3034 (clb$^+$ gifted by Professor Jean-Philippe Nougayrède, France).

HEK cells maintained at standard conditions (37°C / 5% CO2) in DMEM containing L-Glutamine supplemented with 10% FBS and 1% Penicillin/ Streptomycin were used for the cytotoxicity assay. Plates containing HEK cells were incubated for 1 day before the bacteria infection assay cells in DMEM at 37°C, 10% Co2. Overnight culture of clb$^+$ and colibactin negative *E. coli* strains was used for the inoculation of 70–80% confluent cells. Bacteria were suspended in PBS and adjusted OD to 1.0 (2x107 CFU/ml). Bacteria were added accordingly to a multiplicity of infection (MOI) of 10–100 per cell and the plates were incubated at standard conditions for 6 hours. Cells were trypsinized after incubation and trypan blue was mixed with the cell suspension at 1:1. The cell suspension was loaded into a hemocytometer. Live and dead cells were counted under a light microscope as discussed by (Piccinini et al., 2017) [24]. To observe cell morphology, adherent cells were directly exposed to trypan blue at 1:1 and visualized under the light microscope.

## Statistical analysis

The analysis of data was performed using IBM SPSS software. The reliability and validity of data were checked through using Cronbach alpha. The frequencies were calculated, Student's t-test, chi-square, and ANOVA were applied on socio-demographic and presence of colibactin gene data using SPSS, and $p > 0.05$ is considered for the significance of results. Student's t-test and ANOVA were used to calculate significant differences among healthy controls and cancer patients.

## Results and discussion

Colorectal cancer including other severe colon inflammations like ulcerative colitis and the rectal ulcer is a major cause of mortality in Asian countries. Colorectal cancer and colon-associated diseases are commonly associated with dysbiosis of human gut microbiota that leads to colon inflammation leading to colorectal cancer. This disorder may be caused by multiple factors like pathogenic microbes, environmental changes, age, diet, and social behavior including exercise [25]. The current study was designed to characterize pathogenic *E. coli* from colorectal cancer patients visiting the Pakistan Institute of Medical Sciences (PIMS) by exploring their phylogenetic lineages, genotoxin or colibactin producing potential, antibiotic resistance profile, cytotoxic activity, and genetic heterogeneity using cost-effective and rapid ERIC-PCR approach in resource deficient setting.

We screened 60 biopsies collected from individuals who underwent colonoscopy. Histopathological analysis showed that 35 patients were suffering from colorectal cancer and the remaining 25 individuals reported normal. Overall, 75% of the patients included in the colon study were male and 25% were female. Among the male population, 60% of males were suffering from cancer and 40% were normal. Whereas, among the female population 50% of the females were cancerous while 50% of the female population was normal. Chi-square analysis results depict that colorectal cancer is independent of gender ($p = 0.984$). Socio-demographic data analysis showed that hygiene (washing hands after using the toilet and before meals) was significantly associated with colorectal cancer development. People performing frequent hand wash after using toilets are less likely involved in colorectal cancer development ($X_2(1) = 16.09$, $p = 0.001$). Whereas, the people who wash their hands before every meal have fewer chances of CRC development ($X_2(1) = 8.419$, $p = 0.038$). Similarly, patients with frequent intake of potatoes are more likely affected with colorectal cancer ($X_2(1) = 9.49$, $p = 0.023$).

### Phylogroup identification using Quadraplex PCR

Each biopsy was cultured on MacConkey agar plates for detection and isolation of *E. coli*. Suspected *E. coli* colonies were identified using biochemical tests and all forty biopsies were found

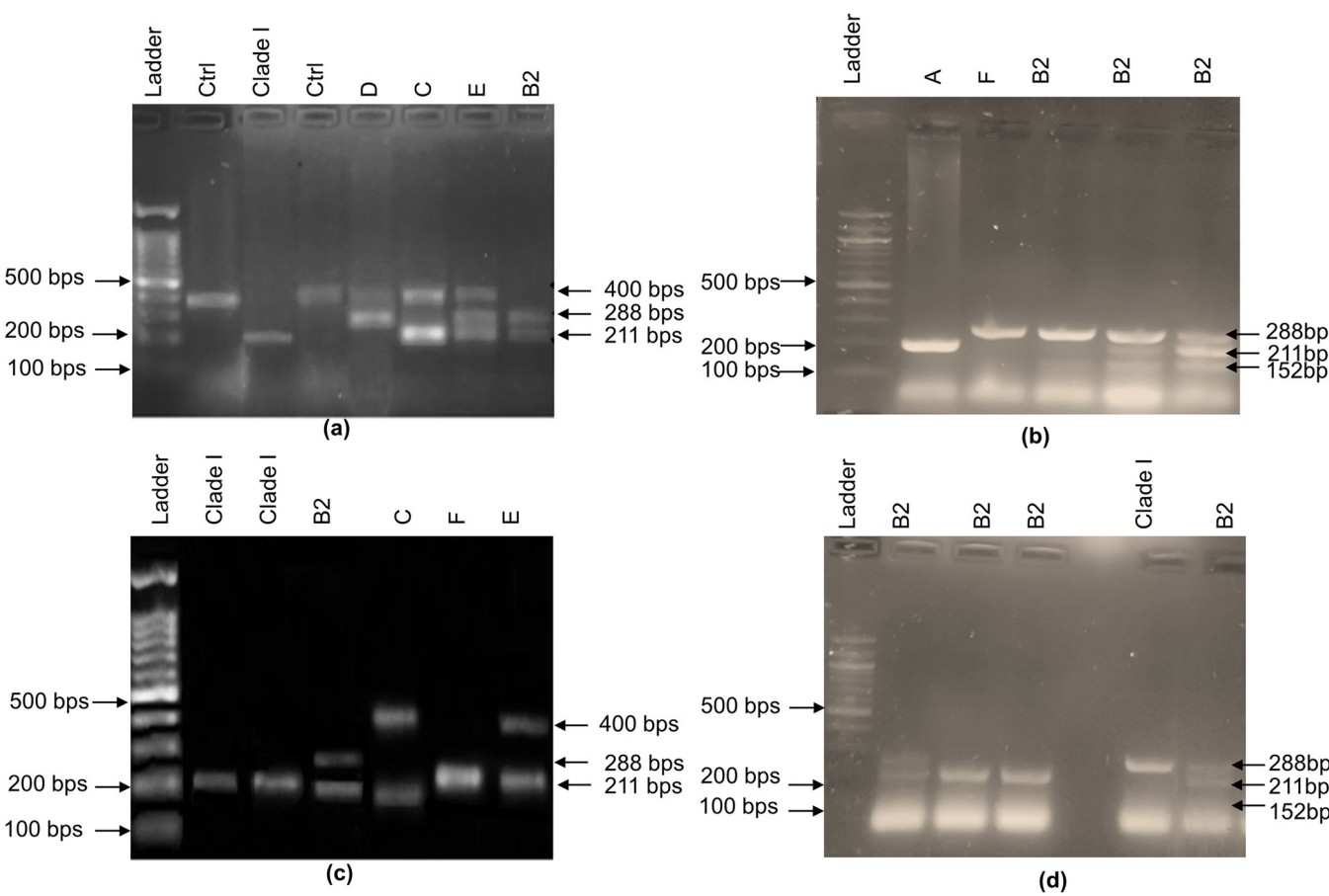

**Fig 1. (a)** Phylogroup PCR related to 35 CRC related *E. coli* strains, well 1 represents 100 bps Gene Ruler DNA ladder, well 2 and 4 represents unknown phylogroup, well 3 represents clade I, while well 5,6, 7, and 8 show phylogroup D, C, E, and B2 phylogroup respectively. **(b)** Well 1 represents 100 bps Gene Ruler DNA ladder, well 2 and 3 represent A and F phylogroups respectively whereas the rest of the wells represent B2 phylogroup. **(c)** Well 1 represents 100 bps Gene Ruler DNA ladder, well 2 and 3 represent clade I/II, well 4,5,6 and 7 represent phylogroups B2, C, F, and E respectively. **(d)** Well 1 represents 100 bps Gene Ruler DNA ladder, well 2, 3,4 and 6 show B2 phylogroup while well 5 shows clade I/II.

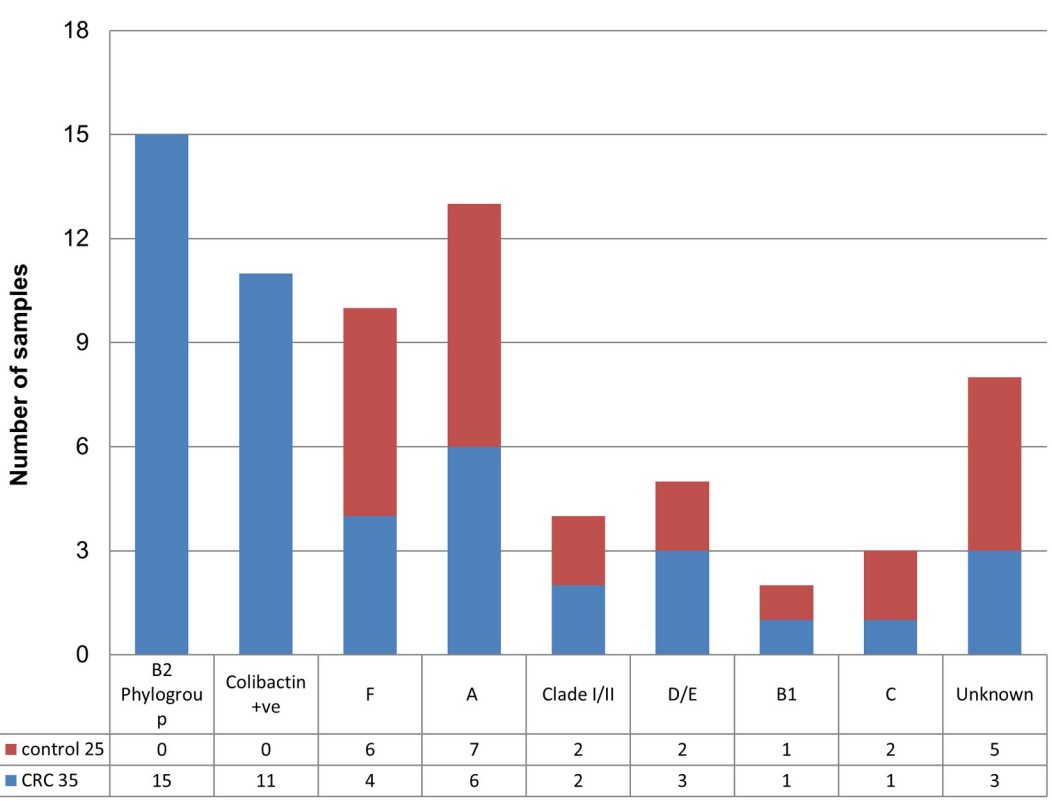

| | B2 Phylogroup | Colibactin +ve | F | A | Clade I/II | D/E | B1 | C | Unknown |
|---|---|---|---|---|---|---|---|---|---|
| ■ control 25 | 0 | 0 | 6 | 7 | 2 | 2 | 1 | 2 | 5 |
| ■ CRC 35 | 15 | 11 | 4 | 6 | 2 | 3 | 1 | 1 | 3 |

**Fig 2. Represents 10 *E. coli* strains belonging to B2 phylogroup isolated from CRC patients. (a)** L presents 1kb GeneRuler DNA Marker, well 2 to 4 presents ClbA 342 bps and ClbQ 308bps bands. **(b)** Well, 1 presents a 1kb GeneRuler ladder, well 2 to well 5 presents ClbA and ClbQ bands while well # 6 contains negative control. **(c)** 1st well represents 100bps GeneRuler DNA ladder, well 2 and 4 shows ClbA, ClbQ, and ClbB 579 bps bands whereas well 3 show a negative result for colibactin genes.

positive for *E. coli* culture. To study the genetic diversity of the *E. coli* isolates phylogenetics analysis was done using primers for phylogroups i.e. chuA, yjaA, TspE4.C2, arpA listed in (S1 Table) (Clermont *et al.*, 2013). Our results showed that 42.85% (15 out of 35) of cancer-associated *E. coli* belong to B2 phylogroup while the rest belonged to other phylogroups i.e., Unknown (3) = A (3) = D/E (3) > Clade I/II (2) > C (1) > B1 (1) (Fig 1). However, *E. coli* strains isolated from the control group referred to phylogroups other than B2 i.e. Unknown (5) > F (3) > A (2) = D/E (2) = Clade I/II (2) = C (2) > B1 (1). So, the majority of the isolates from healthy controls 29% belonged to the unknown phylogroup. According to recent studies, it is now documented that pathogenic strains of *E. coli* majorly belong to B2 and D phylogroup and these strains tend to possess more virulence factors in comparison to commensal strains of *E. coli* [26]. Moreover, B2-phylogroup associated *E. coli* strains harbor *Pks* genomic island that is responsible for the production of non-ribosomally synthesized polyketide-peptide genotoxin known as colibactin [27]. However, we found none of the *E. coli* strains associated with healthy controls belonged to the B2 phylogroup. According to the study reported by Buc and colleagues, among the phylogroups mentioned for *E. coli*, a greater relative abundance belonging to the B2 and D phylogenetic group has been reported in CRC and IBD patients [28]. We surprisingly found different phylogenetic distribution as compared to preceding reports, including a high prevalence of B2 phylogroup among cancer patients but the less and equal frequency of A, D/E, and Unknown phylogroups in cancer samples. On contrary to our study, Zarei *et al.* reported the abundance of phylogroup B2 (30%) and A (52.5%) among colorectal cancer patients [29].

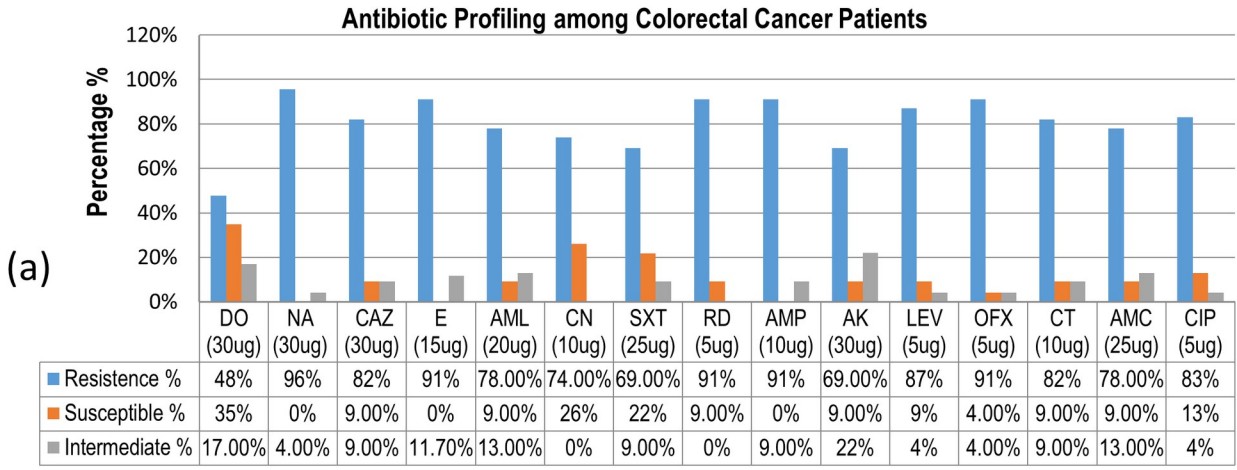

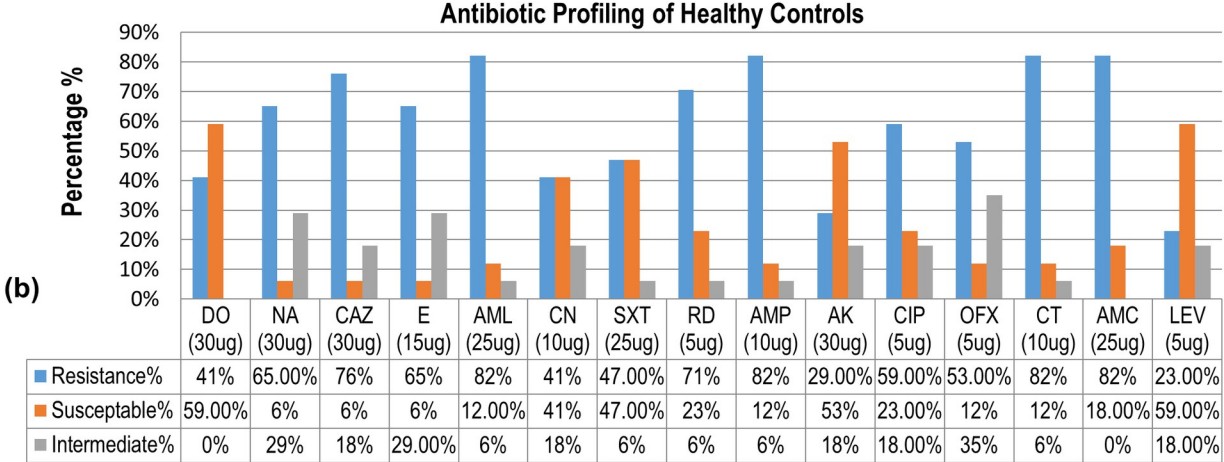

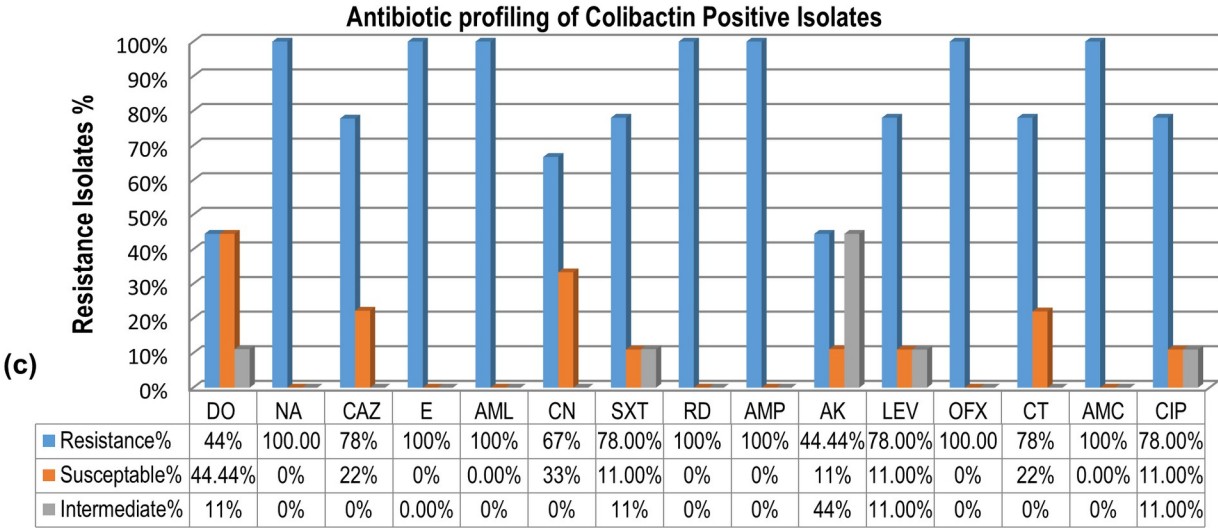

**Fig 3.** (a) Graphical representation for frequency of antibiotic resistance of *E. coli* isolates among colon cancer patients and healthy individuals against each antibiotic tested. (b) Graphical representation of antibiotic profiling against clb+ isolates.

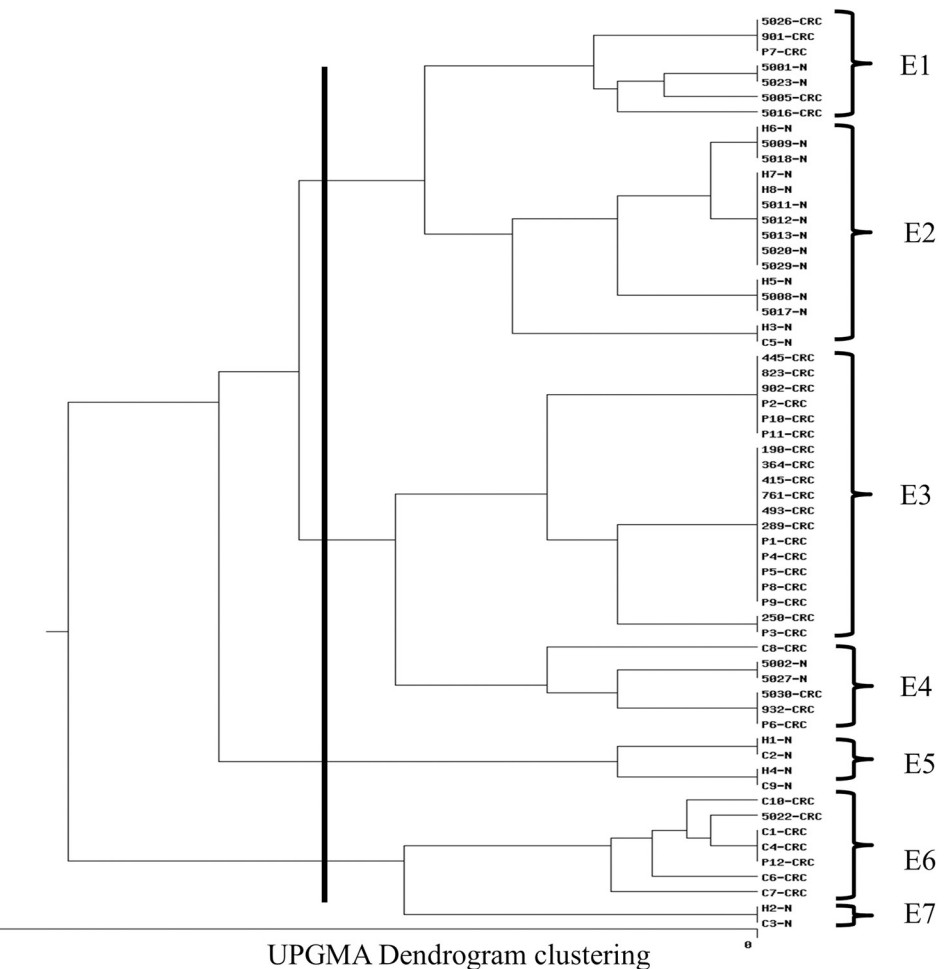

**Fig 4. Dendrogram showing the phylogenetic grouping of the 40 *E. coli* strains studied.**

## Colibactin gene identification

*E. coli* isolates belonging to B2 phylogroups were further investigated for genotoxin (colibactin) production. The primers used for the determination of colibactin-producing *E. coli* isolates are ClbA and ClbQ as in (S1 Table). Our results showed that among 35 cancer patients 15 B2$^+$ (B2 positive) isolates were found as clb$^+$ while from the rest B2$^-$ (B2 negative) cancer-associated isolates none was found positive for colibactin gene as same in healthy controls. Our results showed that 100% of B2 isolates associated with CRC patients were found positive for colibactin genes (Fig 2). This affirms significant $p<0.002$ association of genotoxin producing colibactin gene and colorectal cancer, this association of colorectal cancer and a genotoxin (colibactin) producing *E. coli* is consistent with earlier studies by Martin *et al*. that 70% of colorectal cancer patients possessed mucosa-associated bacteria and from those bacteria, the majority belonged to *E. coli* species [25].

## Antibiotic resistance profiling

The antibiotic susceptibility of *E. coli* isolates was ascertained by standardized methods such as the Kirby-Bauer disk diffusion method on Mueller-Hinton agar plates keeping CLSI 2017 recommendations as standards. *E. coli* strains positive for colibactin showed that more than 60%

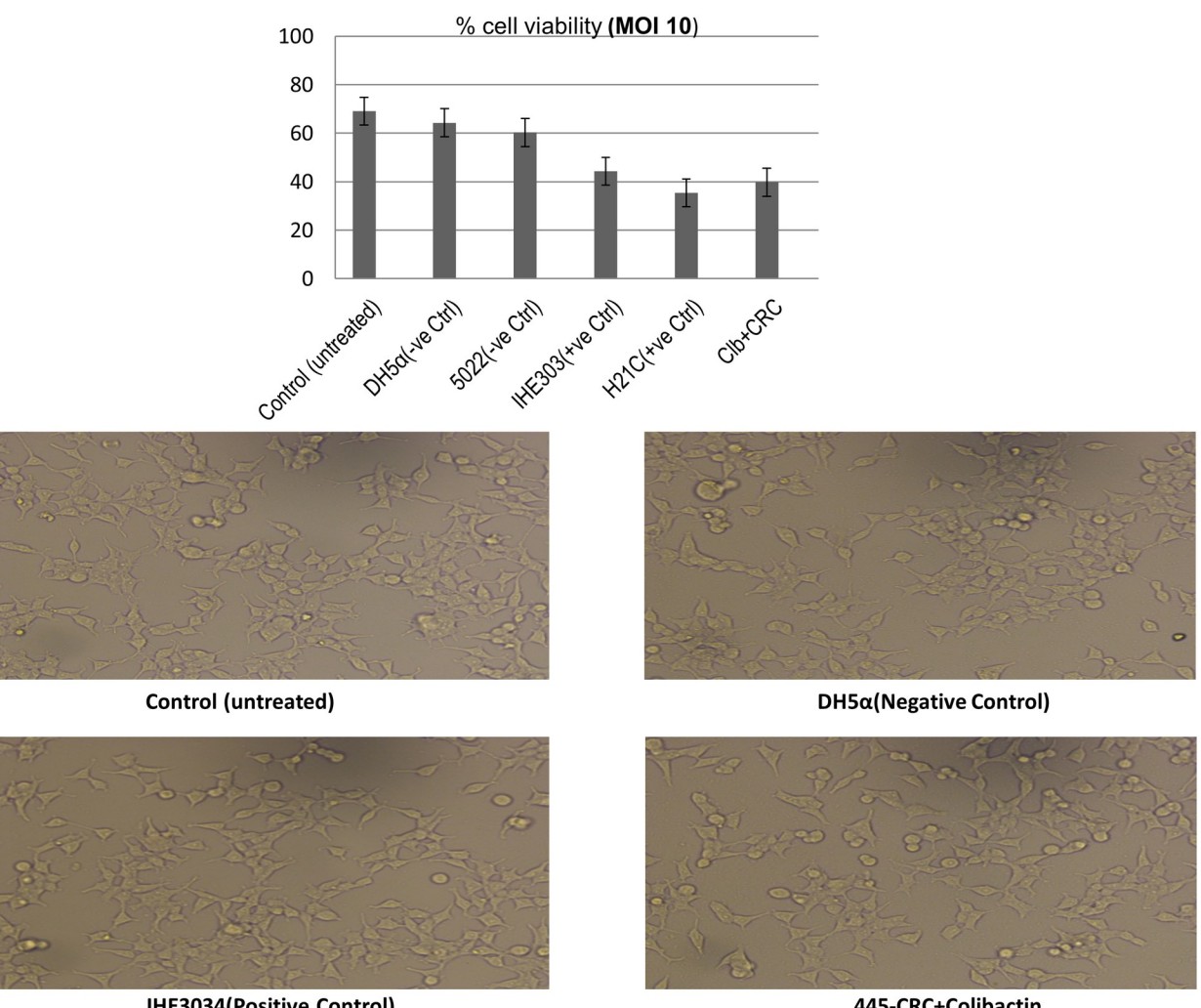

**Fig 5. Cytotoxic activity of clb⁺ *E. coli* strains (190-CRC, 364-CRC) and colibactin negative *E. coli* strain (5002) with standard controls including IHE3034 and H21C.**

of the isolates were resistant to Ampicillin (91%), Rifampin (91%), Ofloxacin (91%), Nalidixic acid (96%), Colistin (82%), Amoxicillin (AMC) (78%), ceftazidime (82%), and Erythromycin (91%) (Fig 3). This study relates to another similar study, that reported high resistance rates of isolated *E. coli* towards ampicillin (77.8%), amoxicillin/clavulanic acid (88.9%), and Nalidixic acid (66.7%) were observed [30].

## ERIC- PCR profiling

To determine the evolutionary relationship among the forty (40) isolates, genotyping was performed using ERIC fingerprinting assay. In this assay, a 22bps primer designed against the specific and conserved ERIC region was used. The PCR was performed in a total volume of 25μl reaction mixture containing 100ng of DNA and 25pmol of ERIC 2 primer (5-AAGTAAGT-GACTGGGGTGAGCG-3) [23]. ERIC-PCR band patterns were calculated using the visibility and their molecular weights in presence of molecular markers. Bands for each sample were counted using the zero-one manual method and then, the data were entered on the following

site: http://insilico.ehu.es/dice_upgma/, and the dendrogram was drawn. Our results corroborate another study that also reported high genetic diversity among *E. coli* isolates extracted from patients suffering from urinary tract infections and presented the dendrogram using the ERIC-PCR method [31]. Phylogenetic analyses using ERIC PCR categorized all isolates in five categories and suggested that CRC associated *E. coli* were found to be genetically diverse when compared using ERIC-PCR (Fig 4).

### In vivo viability/cytotoxicity activity of Clb+ *E. coli* strains

To verify the cytotoxic colibactin activity in the Clb-positive *E. coli* isolates and to study whether clb+ isolates showed enhanced cytotoxicity as compared to strains lacking colibactin, viability assay using HEK cells was done. Our results showed that all the strains irrespective of the phylogroup were cytolethal at MOI 10. At 10 MOI clb$^+$ isolates (control positive as well as isolates) showed significant high cytotoxicity i.e., <95% of the HEK cell were killed as compared to colibactin negative strains (i.e., 85.46% killing). Our results indicate that isolates possessing the clb gene showed cytotoxic activity which may have contributed to the rapid progression of colorectal cancer in patients (Fig 5).

## Conclusions

Cancer-associated *E. coli* isolates were found to be multidrug-resistant and genetically diverse. Moreover, the association of these isolates to phylogroup B2 and the presence of the colibactin gene suggest that their presence in cancer patients may likely aggravate the dysbiosis by altering the normal gut microbiota leading to severe CRC disease. It will be worth checking the production of colibactin in these isolates and also contextualizing them in a global perspective by detailed comparative genomics analysis for a better understanding of their role in CRC.

## Supporting information

**S1 File.**
(XLSX)

**S1 Raw images.**
(TIF)

**S1 Table. Detailed list of primers used in the study [32–34].**
(DOCX)

**S2 Table. Socio-demographic data of patients was collected using questionnaires.**
(DOCX)

## Acknowledgments

The authors acknowledge the patients who volunteered in this study.

## Author Contributions

**Conceptualization:** Habiba Tariq, Habib Bokhari.

**Data curation:** Habiba Tariq, Zobia Noreen, Muhammad Malik.

**Formal analysis:** Habiba Tariq, Zobia Noreen, Aftab Ahmad, Laraib Khan, Mashhood Ali, Aneela Javed, Faisal Rasheed, Alina Fatima, Tanil Kocagoz, Ugur Sezerman.

**Investigation:** Habiba Tariq.

**Methodology:** Zobia Noreen, Laraib Khan, Mashhood Ali, Aneela Javed, Faisal Rasheed, Alina Fatima, Tanil Kocagoz, Ugur Sezerman.

**Project administration:** Zobia Noreen.

**Software:** Aftab Ahmad.

**Supervision:** Habib Bokhari.

**Validation:** Aftab Ahmad, Aneela Javed, Faisal Rasheed, Ugur Sezerman.

**Visualization:** Aftab Ahmad.

**Writing – original draft:** Habiba Tariq.

**Writing – review & editing:** Habiba Tariq, Aftab Ahmad, Laraib Khan, Alina Fatima.

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
