## [Decision Letter · Decision Letter 0]

21 May 2021

PONE-D-21-08856

Colibactin possessing E. coli isolates in association with colorectal cancer and their genetic diversity among Pakistani Population

PLOS ONE

Dear Dr. Bokhari,

Thank you for submitting your manuscript to PLOS ONE. After careful consideration, we feel that it has merit but does not fully meet PLOS ONE’s publication criteria as it currently stands. Therefore, we invite you to submit a revised version of the manuscript that addresses the points raised during the review process.

We look forward to receiving your revised manuscript.

Kind regards,

Grzegorz Woźniakowski, Full professor, PhD, ScD

Academic Editor

PLOS ONE

Journal Requirements:

2.The research project was presented to the ethical review board, PIMS Islamabad. Informed written consent was solicited from enrolled patients.".  

Please amend your current ethics statement to confirm that your named institutional review board or ethics committee specifically approved this study.

Reviewers' comments:

Reviewer's Responses to Questions

**Comments to the Author**

1. Is the manuscript technically sound, and do the data support the conclusions?

Reviewer #1: No

Reviewer #2: Partly

Reviewer #3: Partly

2. Has the statistical analysis been performed appropriately and rigorously? 

Reviewer #1: No

Reviewer #2: I Don't Know

Reviewer #3: I Don't Know

3. Have the authors made all data underlying the findings in their manuscript fully available?

Reviewer #1: No

Reviewer #2: Yes

Reviewer #3: Yes

4. Is the manuscript presented in an intelligible fashion and written in standard English?

Reviewer #1: No

Reviewer #2: No

Reviewer #3: No

5. Review Comments to the Author

Reviewer #1: This study aims to characterize the colibactin production in E. coli isolated from patients with or without colorectal cancer. While this type of descriptive work is important the current study has a number of defects. Firstly, it is of low power: there are only 23 cancer patients and 17 healthy controls. There is little indication of any power analysis to demonstrate why this number was picked and whether any meaningful conclusions can be drawn from such a cohort. Secondly, the agarose gel electrophoresis presented in Figures 1 and 2 are poor quality, presented poorly and the conclusions drawn from them poorly integrated into the manuscript text. This makes it hard to understand what is being concluded from these figures. Thirdly, the reason for the antibiotic resistance profiling is unclear in the broader context of the study. It’s nice to know what the resistance profile is but why include it here? Fourthly, the cytotoxicity data in Figure 5 seems selective. Why look at these strains only? What do the bars represent? How many times have the experiments been performed? Are there meaningful differences between groups? Statements in the results section seem unsupported by data (for example: “Our results showed that all the strains irrespective of the phylogroup were cytolethal at MOI above 50” – where is the data supporting this?). These are just a few examples of the problems with this study which indicate that it is of low quality.

Reviewer #2: Manuscript “Colibactin possessing E. coli isolates in association with colorectal cancer and their genetic diversity among Pakistani Population” is a very relevant study to show the association between specific Colibactin producing strain of E. coli with Colorectal Cancer. It is well designed epidemiological study accounting the dietary habit as well as hygiene practice in low socioeconomic backgrounds. Study mainly focused on strains harboring Pks Island which resides colibactin gene A and Q. Antibiogram profile of the isolates and ERIC-PCR Genotyping were among the tools used to type the strains. Overall, it is an informative study. Followings are my comments:

Majors:

1. As main massage of this paper is to show the association of Colibactin positive E. coli strain with CRC, the data

showed be compared and plotted using proper statistical tools to show this association. May be bar diagram: no.

of CRC isolates vs healthy isolates with and without colibactin gene instead of Fig. 2.

2. It would be informative to analyze individual antibiotic profile with presence of colibactin gene with CRC in Fig. 3.

3. Figure 5 does not have SD or SEM. So, hard to make any conclusion as both colibactin positive E. coli strain as

well as negative control showed toxicity above 85%. Better to use toxicity negative control strain which

showed <10% toxicity when compared with colibactin positive E. coli strain.

Minors:

1. Abstract is too long.

2. Table 1 primer list can be moved to supplement.

3. Figure quality in Fig. 1 and 2 low.

Reviewer #3: The present work describing the “Colibactin possessing E. coli isolates in association with colorectal cancer and their genetic diversity among Pakistani Population” is original and interesting. However, it requires considerable improvements.

General comments:

- Next submission, please, insert pages and lines!

- Abbreviations: define each abbreviation when it first appears in the text. After that, keep using the abbreviation. Sometimes words appear in abbreviated, sometimes in full formats (ex: CRC, pks, etc…) Standardize. Define “PIMS” when it first appears.

- Standardization: For example: for polyketide synthase “–producing”, “-positive” E. coli, I suggest using “pks+ E. coli” throughout the text. The same for colibactin-positive E. coli (clb+ E. coli)

- References: Instead of “XXX and his companions”, use “XXX et al.,”.

- It is Ofloxacin and Nalidixic acid (and not Oflaxacin and Nilidixic acid)

- In “Results and Discussion” session, authors present statistical data and association between hygiene and CRC development. However, there is no mention on how these data have been collected and analyzed. Provide details of methodology used.

- English revision is recommended. Sentence structures and conjunctions are frequently misused.

- Discussion and Conclusions could be better explored.

Minor comments/suggestions:

- Abstract, Line 6: “in the present study, colibactin positive E. coli…”

- Abstract , penultimate sentence: (Suggestion) substitute “populace” with “population”

- Introduction, Paragraph 4: “In 2006, Nougayrède et al. identified a meningitis E. coli strain IHE3034” / The sentence continues “…they named THIS toxin as colibactin”. Strain IHE3034 is not a toxin. Rephrase. // At the end of this paragraph: “… have shown an increased number of tumor in various CRC mouse models”

- Introduction, Paragraph 5: “Recent studies have intimated (implied? suggested?) that some specific E. coli strains possess a cluster of genes named polyketide synthase (pks) island, which has been associated with development of CRC in humans (Arthur et al., 2012)”

- Introduction, Paragraph 6: “HeLa cells”

- Introduction, Paragraph 7: “The isolated E. coli strains…”

- Sample collection: “Patients under 20 year of age suffering from severe (???) infection and bleeding were not included in this study.” Gastrointestinal infection and bleeding? general infection?

- Bacterial culture and Biochemical identification: Confusing. Rewrite. Suggestion: A biopsy sample from each patient was collected, kept in 20% glucose solution and transported to the laboratory on the ice. Biopsies were then homogenized and the preparations were cultured on MacConkey agar plates incubated at 37°C for 18-24 hours for bacterial colonies isolation. Based on morphological characterization, single E. coli colonies were re-streaked and purified on MacConkey agar/LB agar incubated at 37°C overnight. The identification of E. coli colonies were confirmed by Gram’s staining and various biochemical tests (i.e. Triple Sugar Iron (TSI), Citrate Test, Catalase, Oxidase, Indole Test, Methyl red test and Urease test). Pure E. coli cultures confirmed by morphological and biochemical tests were processed and preserved in 20% glycerol and stored at -80 °C until further processing.

- DNA extraction and quantification: Genomic DNA from E. coli strains were extracted by ethanol precipitation method using the method described by (Rasheed et al., 2011). Briefly, a single pure bacterial colony selected from each sample, was resuspended in 20 µL of 1% sodium dodecyl sulfate, 40 µL proteinase K (100 µg/mL), 80 µL of proteinase K buffer (4M NaCl, 0.5M EDTA; pH 7.5), and incubated at 55 °C for 1 hour.

- Phylogrouping: Genomic DNA extracted from each E. coli isolate was used for…. // The primers used for determination of phylogroups (chuA, yjaA, TspE4.C2, and arpA) are listed in Table 1.

- Detection of Colibactin Genes: The primers used for the determination of colibactin producing E.coli isolates (ClbA and ClbQ) are shown in Table 1.

- Antibiotics Susceptibility Profiling: E. coli strains were classified as resistant, susceptible, intermediate by comparison of diameter of inhibition according to the CLSI, 2017 guidelines.

- ERIC-PCR Genotyping: To determine the evolutionary relationship among the 40 E. coli isolates studied, genotyping was performed using ERIC fingerprinting assay.

- Viability/ Cytoxicity of Bacterial Culture against HeLa Cell Lines: Correct “CO2” and “Co2”: CO2 // “Six well plates containing HeLa cells...”. // “Bacteria were suspended in PBS and OD adjusted to 1.0 (2x107 CFU/mL).”

- Results and Discussion, second paragraph: observe comments above. How was these data accessed? How was the hygiene and potato intake accessed? Was the consumption of other food also accessed? // “Whereas, among female population 50% of the females were cancerous while 50% of females appeared normal.” Appeared” (??) vague

- Phylogroup Identification using Quadraplex PCR: “According to recent studies (REF), the majority of pathogenic E. coli strains belong to B2 and D phylogroups, and these strains tend to possess more virulence factors in comparison to commensal strains of E. coli.”

- Colibactin Gene Identification: “Our results showed that nine (9) out of ten (10/23) B2 phylogroup cancer-associated isolates were found as colibactin possessing E. coli.” Rewrite this sentence to make it clearer. “9 out of 10… (10/23)” is a bit confusing… // It would be easier if authors use “clb+” or “clb- E.coli” notation. It is important to observe that it is not the patient or heathy individual the one who “has positive/negative colibactin gene, but the E. coli strains. “Seventeen healthy individuals presented clb- E. coli strains, while among 23 CRC patients, 14 and 9 respectively presented clb- and clb+ E. coli strains.”

- In vivo viability/cytotoxicity activity of Colibactin Positive E. coli Strains: “Our results showed that ALL the strains irrespective of the phylogroup were cytolethal at MOI above 50.” But, as described in Material and Methods, this “all” strains refers to only 2 clb+ and 1 clb- E.coli (and 2 positive controls). This must be clear in this paragraph. Have you checked if these strains also produce other cyclomodulins like CDT and CNF? I think that a cytotoxin negative control should have been included in this experiment.// Revise: “(2 positive controls and 2 E. coli strains isolated in this study)”

- Table 1: The following primers are missing: ClbQ R, ClbB F and R.

- Figure 1, legend: Clearly state that this figure refers to the 23 CRC related E. coli strains.

- Figure 2, legend: Clearly state that this figure refers to the 10 E. coli strains belonging to the B2 phylogroup which were isolated from CRC cases.

Figure 3: “Graphical representation for antibiotic profiling of E. coli isolates among colon cancer patients and healthy individuals”. This not represents antibiotic profile of E. coli. This represents the frequency of antibiotic resistance of this group of E. coli for each antibiotic tested.

- Figure 4: “Dendrogram showing the phylogenetic grouping” of the 40 E. coli strains studied.

- Conclusions: Authors concluded that “Cancer-associated E. coli isolates were found to be multidrug-resistant and genetically diverse.” Would it be the equivalent to say that “clb+ E.coli are multidrug-resistant and genetically diverse”? Regarding antibiotic resistance experiment, the only result presented was the frequency of resistant isolates for individual antibiotics. Would be very interesting to compare resistance profiles presented by clb+ and clb- E. coli groups, and see if clb+ strains are in fact more “multidrug”-resistant than clb- strains. Likewise, it would be interesting to see the distribution of both groups in dendrogram presented in Figure 4.

6. PLOS authors have the option to publish the peer review history of their article (what does this mean?). If published, this will include your full peer review and any attached files.

Reviewer #1: No

Reviewer #2: No

Reviewer #3: No

---

## [Author Response · Author response to Decision Letter 0]

15 Nov 2021

Response to reviewers

With Reference ; 

PONE-D- 21-08856

“Colibactin possessing E. coli isolates in association with colorectal cancer and their genetic diversity among Pakistani Population”.

First of all, we would like to thank reviewers for their valuable comments and time. 

The specific comments and their responses are mentioned below.

Reviewer #1: This study aims to characterize the colibactin production in E. coli isolated from patients with or without colorectal cancer. While this type of descriptive work is important the current study has a number of defects.

Firstly, it is of low power: there are only 23 cancer patients and 17 healthy controls. There is little indication of any power analysis to demonstrate why this number was picked and whether any meaningful conclusions can be drawn from such a cohort.

Secondly, the agarose gel electrophoresis presented in Figures 1 and 2 are poor quality, presented poorly and the conclusions drawn from them poorly integrated into the manuscript text. This makes it hard to understand what is being concluded from these figures.

Response: Thank you for your suggestion, we have added more samples to increase number up to 60 samples in total. Secondly, agarose gel pictures quality have been improved. Figure 2 has been replaced with a bar graph diagram as per reviewer’s suggestions to present detailed breakdown of samples based on Phylogroups and presence/ absence of colibactin gene. 

Thirdly, the reason for the antibiotic resistance profiling is unclear in the broader context of the study. It’s nice to know what the resistance profile is but why include it here?

Response: We performed antibiotic resistance profiling to report the main cause of reduction in effectiveness of drugs during the treatment of various diseases including colorectal cancer. Drug resistance develops in approximately all colon cancer patients and that leads to decreased therapeutic efficacies of anticancer agents. This is the main reason to analyze resistance patterns among CRC and Healthy individuals. 

Fourthly, the cytotoxicity data in Figure 5 seems selective. Why look at these strains only? What do the bars represent? How many times have the experiments been performed? Are there meaningful differences between groups? Statements in the results section seem unsupported by data (for example: “Our results showed that all the strains irrespective of the phylogroup were cytolethal at MOI above 50” – where is the data supporting this?). These are just a few examples of the problems with this study which indicate that it is of low quality.

Response: Cytotoxicity assay has been repeated as suggested by reviewer and new figure has been added to present the precisely results in effective way. 

Reviewer #2: Manuscript “Colibactin possessing E. coli isolates in association with colorectal cancer and their genetic diversity among Pakistani Population” is a very relevant study to show the association between specific Colibactin producing strain of E. coli with Colorectal Cancer. It is well designed epidemiological study accounting the dietary habit as well as hygiene practice in low socioeconomic backgrounds. Study mainly focused on strains harboring Pks Island which resides colibactin gene A and Q. Antibiogram profile of the isolates and ERIC-PCR Genotyping were among the tools used to type the strains. Overall, it is an informative study. Followings are my comments:

1. As main message of this paper is to show the association of Colibactin positive E. coli strain with CRC, the data showed be compared and plotted using proper statistical tools to show this association. May be bar diagram: no. of CRC isolates vs healthy isolates with and without colibactin gene instead of Fig. 2.

Response: Bar diagram has been added to represent phylogroups with presence or absence of colibactin gene among all cancer and healthy individuals. 

2. It would be informative to analyze individual antibiotic profile with presence of colibactin gene with CRC in Fig. 3.

Response: yes agreed, Bar graph (Fig. 3) has been added to represents the strong association of colibactin gene with CRC.

3. Figure 5 does not have SD or SEM. So, hard to make any conclusion as both colibactin positive E. coli strain as well as negative control showed toxicity above 85%. Better to use toxicity negative control strain which showed <10% toxicity when compared with colibactin positive E. coli strain.

Response: Yes thank you for your worthy suggestions, we have repeated cytotoxicity assay with increased data set including positive and negative controls to address the comments and new figure has been added to present the precisely results in effective way.

Minors:

1. Abstract is too long.

Response: Abstract has been summarized to 300 words according to Plos ONE guide lines regarding abstract. 

2. Table 1 primer list can be moved to supplement.

Response: Primer list has been moved to supporting information as Table S1.

3. Figure quality in Fig. 1 and 2 low.

Response: Figure quality has been improved upto 300dpi.

Reviewer #3: The present work describing the “Colibactin possessing E. coli isolates in association with colorectal cancer and their genetic diversity among Pakistani Population” is original and interesting. However, it requires considerable improvements.

General comments:

- Next submission, please, insert pages and lines!

Response: yes pages and lines have been added. Thank you

- Abbreviations: define each abbreviation when it first appears in the text. After that, keep using the abbreviation. Sometimes words appear in abbreviated, sometimes in full formats (ex: CRC, pks, etc…) Standardize. Define “PIMS” when it first appears.

Response: All required abbreviation have been defined.

- Standardization: For example: for polyketide synthase “–producing”, “-positive” E. coli, I suggest using “pks+ E. coli” throughout the text. The same for colibactin-positive E. coli (clb+ E. coli)

Response: Thank you, yes it has been corrected accordingly.

- References: Instead of “XXX and his companions”, use “XXX et al.,”.

Response: Yes changes have been made accordingly.

- It is Ofloxacin and Nalidixic acid (and not Oflaxacin and Nilidixic acid)

Response: yes it has been corrected, thank you.

- In “Results and Discussion” session, authors present statistical data and association between hygiene and CRC development. However, there is no mention on how these data have been collected and analyzed. Provide details of methodology used.

Response: Socio-demographic data was collected through questionnaire, attached as a supplementary table S2.

- English revision is recommended. Sentence structures a and conjunctions are frequently misused.

Response: We have revisited the complete manuscript considering these comments. There are number of sentences which have been rephrased for better expression. Moreover, language and grammar was thoroughly checked from abstract to conclusions sections and all the necessary technical expression have been mended accordingly. There are number of corrections incorporated in reference list as well. Language and grammar has been revised by 

Habib Bokhari, PhD

Commonwealth Scholar & Fellow (LSHTM, UK)

Fulbright Fellow (Perelman School of Medicine, UPENN, USA)

Professor, Department of Biosciences COMSATS University, Islamabad, Pakistan

Office: +92-51-250-1223; FAX: +92-51-444-2805 Mobile: +92-300-512-7684

E-mail: habib@comsats.edu.pk

- Discussion and Conclusions could be better explored.

Response: Yes it has been improved to improve the expression of manuscript.

Minor comments/suggestions:

- Abstract, Line 6: “in the present study, colibactin positive E. coli…”

Response: Yes it has been fixed.

- Abstract , penultimate sentence: (Suggestion) substitute “populace” with “population”

Response: Yes it has been corrected.

- Introduction, Paragraph 4: “In 2006, Nougayrède et al. identified a meningitis E. coli strain IHE3034” / The sentence continues “…they named THIS toxin as colibactin”. Strain IHE3034 is not a toxin. Rephrase. // At the end of this paragraph: “… have shown an increased number of tumor in various CRC mouse models”

Response: done and corrected

- Introduction, Paragraph 5: “Recent studies have intimated (implied? suggested?) that some specific E. coli strains possess a cluster of genes named polyketide synthase (pks) island, which has been associated with development of CRC in humans (Arthur et al., 2012)”

Response: yes it has been corrected.

- Introduction, Paragraph 6: “HEK, cells”

Response: yes relevant information has been added

- Introduction, Paragraph 7: “The isolated E. coli strains…”

Response: Yes it has been corrected

- Sample collection: “Patients under 20 year of age suffering from severe (???) infection and bleeding were not included in this study.” Gastrointestinal infection and bleeding? general infection?

Response: Yes it was a typos mistake it has been fixed.

- Bacterial culture and Biochemical identification: Confusing. Rewrite. Suggestion: A biopsy sample from each patient was collected, kept in 20% glucose solution and transported to the laboratory on the ice. Biopsies were then homogenized and the preparations were cultured on MacConkey agar plates incubated at 37°C for 18-24 hours for bacterial colonies isolation. Based on morphological characterization, single E. coli colonies were re-streaked and purified on MacConkey agar/LB agar incubated at 37°C overnight. The identification of E. coli colonies were confirmed by Gram’s staining and various biochemical tests (i.e. Triple Sugar Iron (TSI), Citrate Test, Catalase, Oxidase, Indole Test, Methyl red test and Urease test). Pure E. coli cultures confirmed by morphological and biochemical tests were processed and preserved in 20% glycerol and stored at -80 °C until further processing.

Response: yes it has been corrected

- DNA extraction and quantification: Genomic DNA from E. coli strains were extracted by ethanol precipitation method using the method described by (Rasheed et al., 2011). Briefly, a single pure bacterial colony selected from each sample, was resuspended in 20 μL of 1% sodium dodecyl sulfate, 40 μL proteinase K (100 μg/mL), 80 μL of proteinase K buffer (4M NaCl, 0.5M EDTA; pH 7.5), and incubated at 55 °C for 1 hour.

Response: yes it has been corrected, thank you

- Phylogrouping: Genomic DNA extracted from each E. coli isolate was used for…. // The primers used for determination of phylogroups (chuA, yjaA, TspE4.C2, and arpA) are listed in Table 1.

Response: yes it has been corrected, thank you 

- Detection of Colibactin Genes: The primers used for the determination of colibactin producing E.coli isolates (ClbA and ClbQ) are shown in Table 1.

Response: yes it has been corrected, thank you 

- Antibiotics Susceptibility Profiling: E. coli strains were classified as resistant, susceptible, intermediate by comparison of diameter of inhibition according to the CLSI, 2017 guidelines.

Response: yes it has been corrected, thank you 

- Viability/ Cytoxicity of Bacterial Culture against HeLa Cell Lines: Correct “CO2” and “Co2”: CO2 // “Six well plates containing HeLa cells...”. // “Bacteria were suspended in PBS and OD adjusted to 1.0 (2x107 CFU/mL).”

Response: Yes cytotoxicity assay has been repaeted with HEK cell lines and correction has been made accordingly. Thank you

- Results and Discussion, second paragraph: observe comments above. How was these data accessed? How was the hygiene and potato intake accessed? Was the consumption of other food also accessed? // “Whereas, among female population 50% of the females were cancerous while 50% of females appeared normal” Appeared” (??) vague

Response: Socio-demographic data was collected through questionnaire, attached as a supplementary table S2.

- Phylogroup Identification using Quadraplex PCR: “According to recent studies (REF), the majority of pathogenic E. coli strains belong to B2 and D phylogroups, and these strains tend to possess more virulence factors in comparison to commensal strains of E. coli.”

Response: yes it has been corrected

- Colibactin Gene Identification: “Our results showed that nine (9) out of ten (10/23) B2 phylogroup cancer-associated isolates were found as colibactin possessing E. coli.” Rewrite this sentence to make it clearer. “9 out of 10… (10/23)” is a bit confusing… // It would be easier if authors use “clb+” or “clb- E.coli” notation. It is important to observe that it is not the patient or heathy individual the one who “has positive/negative colibactin gene, but the E. coli strains. “Seventeen healthy individuals presented clb- E. coli strains, while among 23 CRC patients, 14 and 9 respectively presented clb- and clb+ E. coli strains.”

Response: it has been corrected

- In vivo viability/cytotoxicity activity of Colibactin Positive E. coli Strains: “Our results showed that ALL the strains irrespective of the phylogroup were cytolethal at MOI above 50.” But, as described in Material and Methods, this “all” strains refers to only 2 clb+ and 1 clb- E.coli (and 2 positive controls). This must be clear in this paragraph. Have you checked if these strains also produce other cyclomodulins like CDT and CNF? I think that a cytotoxin negative control should have been included in this experiment.// Revise: “(2 positive controls and 2 E. coli strains isolated in this study)”

Response: thank you , agreed, DH5-alpha has been added as cytotoxin negative control 

- Table 1: The following primers are missing: ClbQ R, ClbB F and R.

Response: Yes fine, misssing details has been added. Thank you

- Figure 1, legend: Clearly state that this figure refers to the 23 CRC related E. coli strains.

Response: Yes, it has been fixed. 

- Figure 2, legend: Clearly state that this figure refers to the 10 E. coli strains belonging to the B2 phylogroup which were isolated from CRC cases.

Response: Yes, it has been corrected.

-Figure 3: “Graphical representation for antibiotic profiling of E. coli isolates among colon cancer patients and healthy individuals”. This not represents antibiotic profile of E. coli. This represents the frequency of antibiotic resistance of this group of E. coli for each antibiotic tested.

Response: Yes, it has been corrected.

- Figure 4: “Dendrogram showing the phylogenetic grouping” of the 40 E. coli strains studied.

Response: yes it has been corrected and updated.

- Conclusions: Authors concluded that “Cancer-associated E. coli isolates were found to be multidrug-resistant and genetically diverse.” Would it be the equivalent to say that “clb+ E.coli are multidrug-resistant and genetically diverse”? Regarding antibiotic resistance experiment, the only result presented was the frequency of resistant isolates for individual antibiotics. Would be very interesting to compare resistance profiles presented by clb+ and clb- E. coli groups, and see if clb+ strains are in fact more “multidrug”-resistant than clb- strains. Likewise, it would be interesting to see the distribution of both groups in dendrogram presented in Figure 4.

Response: Yes, suggested changes have been made according

---

## [Editor Report · Decision Letter 1]

4 Jan 2022

Colibactin possessing E. coli isolates in association with colorectal cancer and their genetic diversity among Pakistani Population

PONE-D-21-08856R1

Dear Dr. Bokhari,

We’re pleased to inform you that your manuscript has been judged scientifically suitable for publication and will be formally accepted for publication once it meets all outstanding technical requirements.

Kind regards,

Grzegorz Woźniakowski, Full professor, PhD, ScD

Academic Editor

PLOS ONE
---

## [Editor Report · Acceptance letter]

14 Oct 2022

PONE-D-21-08856R1 

Colibactin possessing E. coli isolates in association with colorectal cancer and their genetic diversity among Pakistani Population 

Dear Dr. Bokhari:

I'm pleased to inform you that your manuscript has been deemed suitable for publication in PLOS ONE. Congratulations! Your manuscript is now with our production department. 

Kind regards, 

on behalf of

Prof. Grzegorz Woźniakowski 

Academic Editor

PLOS ONE